

# Ligninolytic activity of the *Penicillium chrysogenum* and *Pleurotus ostreatus* fungi involved in the biotransformation of synthetic multi-walled carbon nanotubes modify its toxicity

Gladys Juárez-Cisneros[1], Jesús Campos-García[2],
Sharel Pamela Díaz-Pérez[2], Javier Lara-Romero[3],
Dhirendra Kumar Tiwari[4], Juan Manuel Sánchez-Yáñez[1],
Homero Reyes-De la Cruz[5], Sergio Jiménez-Sandoval[6] and
Javier Villegas[1]

[1] Laboratorio de Interacción Suelo Planta Microorganismo, Instituto de Investigaciones Químico Biológicas, Universidad Michoacana de San Nicolás de Hidalgo, Morelia, Michoacán, Mexico
[2] Laboratorio de Biotecnología Microbiana, Instituto de Investigaciones Químico Biológicas, Universidad Michoacana de San Nicolás de Hidalgo, Morelia, Michoacán, Mexico
[3] Facultad de Ingeniería Química, Universidad Michoacana de San Nicolás de Hidalgo, Morelia, Michoacán, Mexico
[4] El Colegio de Michoacán, Ladipa, La Piedad, La Piedad, Michoacán, Mexico
[5] Laboratorio de Control Traduccional, Instituto de Investigaciones Químico Biológicas, Universidad Michoacana de San Nicolás de Hidalgo, Morelia, Michoacán, Mexico
[6] Unidad Querétaro, Centro de Investigación y de Estudios Avanzados del IPN, Querétaro, Querétaro, Mexico

Corresponding authors
Jesús Campos-García,
jesus.campos@umich.mx
Javier Villegas,
vilj4455@yahoo.com.mx

## ABSTRACT

Multi-walled carbon nanotubes (MWCNTs) are of multidisciplinary scientific interest due to their exceptional physicochemical properties and a broad range of applications. However, they are considered potentially toxic nanoparticles when they accumulate in the environment. Given their ability to oxidize resistant polymers, mycorremediation with lignocellulolytic fungi are suggested as biological alternatives to the mineralization of MWCNTs. Hence, this study involves the ability of two fungi specie to MWCNTs biotransformation by laccase and peroxidases induction and evaluation in vivo of its toxicity using *Caenorhabditis elegans* worms as a model. Results showed that the fungi *Penicillium chrysogenum* and *Pleurotus ostreatus* were capable to grow on media with MWCNTs supplemented with glucose or lignin. Activities of lignin-peroxidase, manganese-peroxidase, and laccase in cultures of both fungi were induced by MWCNTs. Raman, FTIR spectroscopy, HR-TEM, and TGA analyses of the residue from the cultures of both fungi revealed structural modifications on the surface of MWCNTs and its amount diminished, correlating the MWCNTs structural modifications with the laccase-peroxidase activities in the fungal cultures. Results indicate that the degree of toxicity of MWCNTs on the *C. elegans* model was enhanced by the structure modification associated with the fungal ligninolytic activity. The toxic effect of MWCNTs on the in vivo model of worms reveals the increment of reactive oxygen species as a mechanism of toxicity. Findings indicate that the MWCNTs can be subject in nature to biotransformation

processes such as the fungal metabolism, which contribute to modify their toxicity properties on susceptible organisms and contributing to environmental elimination.

## INTRODUCTION

Carbon nanotubes (CNTs) are nanomaterials that consist of one or several sheets of crystalline carbon rolled in concentric cylinders (*Saifuddin, Raziah & Junizah, 2013*). They are of scientific, commercial, and environmental interest due to their unique mechanical, electrical, and optical properties (*Zaytseva, 2016*). Specifically, CNTs are increasingly used in areas such as biomedical, genetics, electronics, cosmetics, pharmacology, and agriculture (*Burlaka et al., 2015*; *De Volder et al., 2013*). However, given the high stability of CNTs, they can accumulate in the ecosystem and lead to environmental and health risks (*Shin, Song & Um, 2015*). Reports suggest that CNTs nanoparticles by their diverse chemical structures, shape, size, and low solubility are potentially toxic for diverse organisms such as microorganisms (*Ge et al., 2016*; *Shin, Song & Um, 2015*; *Uo et al., 2011*; *Yadav, Mungray & Mungray, 2016*), plants (*Rong et al., 2018*; *Lin et al., 2009*; *Tan, Lin & Fugetsu, 2009*), and animals (*Bhattacharya et al., 2013*; *Petersen et al., 2011*). In this context, recent studies focused on determining biological alternatives to degrade or mineralize CNTs to avoid their possible health risks that have been described (*Bhattacharya et al., 2013*; *Petersen, Huang & Weber, 2008*).

In the last decade, in vitro studies report that oxidative enzymes from human, microbial, or plant can catalyze CNT degradation (*Modugno et al., 2016*). Additionally, peroxidase isolated from *Armoracia rusticana* in presence of $H_2O_2$ or $FeCl_3$ partially degrades CNTs structures of a single thick layer or single-walled carbon nanotubes (SWCNTs) (*Allen et al., 2009*; *Magrez et al., 2006*) and multi-walled carbon nanotubes (MWCNTs) (*Petersen, Huang & Weber, 2008*; *Zhao, Allen & Star, 2012*). Similarly, reports indicate that SWCNTs are susceptible to degradation by enzymes from animal cells such as myeloperoxidase (*Kagan et al., 2010*; *Vlasova et al., 2011*), lactoperoxidase (*Bhattacharya et al., 2013*), and eosinophil-peroxidase (*Andón et al., 2013*). Research suggest that mycorremediation by basidiomycetes and deuteromycetes represents a biological alternative to alleviate the accumulation of carbon nanomaterials such as MWCNTs through extracellular peroxidases, such as manganese-peroxidase (MnP), lignin-peroxidase (LiP), or laccase, which exhibit the potential to oxidize MWCNTs (*Fujii et al., 2013*; *Janusz et al., 2013*); through unspecific enzymes that catalyze the oxidation of compounds such as polycyclic aromatic hydrocarbons, phenols, organophosphates, and pesticides (*Gan, Lau & Ng, 2009*; *Haritash & Kaushik, 2009*; *Ting et al., 2011*). Phenol-oxidases such as laccase catalyzes the oxidation, decarboxylation or methoxylation of a wide spectrum of phenolic compounds, aromatic amines, and polynucleoaromatic hydrocarbons. While LiP are relatively nonspecific enzymes for substrates, these can

oxidize aromatic compounds with high redox potential such as veratryl alcohol, methoxybenzenes, and nonphenolic groups of the lignin; in contrast, the MnP are specific by substrates (*Qu, Alvarez & Li, 2013*). In this context, compounds such as pyrene, benzo (a) pyrene, acenaphthene, phenanthrene, anthracene, and fluoranthene can be oxidized by enzymes such as LiP, MnP, and laccases (*Barr & Aust, 1994*; *Wang et al., 2009*).

On the other hand, it has been documented that white rot fungi through an enzymatic mechanism by laccases and MnP excreted by *P. chrysosporiume* transformed and oxidized pristine MWCNT by modifying functional groups of the carbon structure (C=O and O–H radicals), leading to shortening of MWCNTs structure (*Barr & Aust, 1994*; *Ma et al., 2019*). Enzymes from *Sparassis latifolia* can be utilized to biotransform SWCNTs into compounds with lower toxicity and used for mineralization of carbon nanomaterials (*Chandrasekaran et al., 2014*), also as peroxidases and laccases from *Trametes versicolor* and *Phlebia tremellosa* that partially degrades SWCNTs (*Zhang, Chen & Alvarez, 2014*). Thus, fungal enzymes are suggested as candidates to biotransform or biodegrade carbon nanomaterials into simpler and susceptible compounds and represent a biological alternative to control the persistence of nanoparticles in the environment (*Ge et al., 2016*; *Parks et al., 2015*). However, to date, there is a paucity of studies that establish the roles played by microbial communities in the biotransformation of MWCNTs. This is a key point because fungi are the main recyclers of organic material in nature. *Caenorhabditis elegans* is a saprophytic nematode that lives in the liquid phase of the soil and leaf-litter environments, it has been widely utilized as an in vivo animal model in toxicology to evaluate the compounds toxicity such as carbon-based nanomaterials among others (*Chen, Hsiao & Chou, 2013*). Therefore, the objective of the present study involved an in vitro evaluation of the capacity of two fungi, *Pleurotus ostreatus* and *Penicillium chrysogenum* to biotransformation/biodegradation of synthetic MWCNTs, thereby also evaluating its toxicity in *C. elegans* worms as in vivo soil-based model.

# MATERIALS AND METHODS

## MWCNT specifications

The MWCNTs used in the study exhibit the following characteristics: Outer diameter of 6–13 nm, internal diameter of 2.0–4.0 nm × 2.5–20 µm length, average wall thickness 7–13 graphene layers, and purity > 98% (Aldrich, Cat 698849).

## Culture conditions

In the study, *Pleurotus ostreatus* and *Penicillum chrysogenum* were provided from the collection of microorganisms in our laboratory (microscopic morphology is showed in Fig. S1, Supplemental Material). The initial inoculum was prepared by culturing each fungus in a potato dextrose agar medium (PDA; BD-Bioxon) for an incubation period of 7 d at 22 °C. Subsequently, discs of the agar plate corresponding to approximately 1 cm$^2$ for each fungus were inoculated in a 100-mL liquid medium with the following composition (g/L): $NH_4NO_3$ 5.0, $CuSO_4$ 0.01, $MgSO_4$ 1.5, $KH_2PO_4$ 1.5, $K_2HPO_4$ 1.5, NaCl 0.9, yeast extract 1.0, bromothymol blue 10 ppm, and 1 mL/L of trace elements solution

prepared as follows (g/L): $H_3BO_3$ 2.86, $ZnSO_4–7H_2O$ 0.22, $MnCl_2–7H_2O$ 1.81, and $KMnO_4$ 0.09. The carbon source used corresponded to 0.5 g/L glucose and/or 2 g/L of residual lignin from semi-purified wheat straw based on *Baltierra-Trejo et al. (2016)* and 10 µg/mL of MWCNTs at a pH of 5.5. The fungi cultures were incubated on a rotary shaker at 150 rpm for 28 d at 30 °C.

## Determination of enzymatic activities

Specifically, 10-mL samples were collected at intervals corresponding to 7 days from the culture media of *P. chrysogenum* and *P. ostreatus* fungi and enzymatic activities were determined in supernatants. Supernatants were obtained by centrifugation at $8000 \times g$ (SL 40R; Thermo Fisher, Waltham, MA, USA) for 15 min at 4 °C and, thereby eliminating mycelium and residual lignin. The lignin-peroxidase activity was determined via the oxidation of veratryl alcohol to veratraldehyde by measuring absorbance at 310 nm using a spectrophotometer (BioTek Instrument Inc-Epoch, Winooski, VT, USA), the molar extinction coefficient ($\varepsilon$) 9,300 $M^{-1}cm^{-1}$. The reaction mixture was prepared with 400 µL of supernatant on 2.5 mL of sodium acetate buffer, 100 mM, pH 3.0, 1.0 mL of veratryl alcohol (Sigma) 10 mM, and 100 µL of $H_2O_2$ 10 mM. The manganese-peroxidase activity was determined by the oxidation of $MnSO_4$ to tartrate by measuring absorbance at 240 nm using a spectrophotometer, $\varepsilon$ 6,500 $M^{-1} cm^{-1}$ (*Gao et al., 2011*). The reaction mixture was prepared with 400 µL of supernatant, 3.4 mL of sodium malonate buffer (Sigma) 50 mM, pH 4.5, 0.1 mL $MnSO_4$ 15 mM, 100 µL $H_2O_2$ 10 mM. Laccase activity was determined via the oxidation of 2,2′-acino-bis-(3-ethyl-benzothiazoline-6-sulfonic acid) (ABTS) by measuring absorbance at 420 nm using a spectrophotometer; $\varepsilon$ of oxidized ABTS was 36,500 $M^{-1} cm^{-1}$. The reaction mixture was prepared with 300 µL of supernatant, 2.4 mL of sodium acetate buffer 25 mM, pH 3.0, and 300 µL of ABTS 10 mM (*Ibrahim et al., 2011*; *Palmieri et al., 1997*).

## Analyses of MWCNTs

The content and chemical modifications of MWCNTs was analyzed at the beginning (0 days) and end (28 days) of the assay in the fungi cultures (5-mL by sample), which samples were dried using an oven at 72 °C for 48 h and later incinerated at 700 °C for 5 h to mineralize non-crystalline organic matter. Subsequently, the ashes obtained were pulverized and suspended in distilled water or used directly for further qualitative and quantitative analyses.

The samples (0.5 mg of dry sample) analyzed by Raman spectroscopy were dispersed in 50 µL water, aliquots of 10 µL were placed on slides, five drops for each sample, allowed to dry at room temperature to be analyzed in Raman spectrometer (Lab Ram HR Evolution; Horiba, Kyoto, Japan) with a spectral resolution of 5 $cm^{-1}$ using a 20-mW He-Ne laser at 632.8 nm visual excitation, and the spectral range was scanned from 110 $cm^{-1}$ to 3,690 $cm^{-1}$ with an integration time of 5 s (*Lara-Romero et al., 2017*).

Attenuated total reflection Fourier transform infrared spectroscopy of samples obtained under the aforementioned conditions were analyzed, and the spectra were recorded via a

Nicolet iS10 spectrophotometer (Thermo Scientific, Waltham, MA, USA) by the attenuated total reflection technique.

Thermogravimetric (TGA) analysis was carried out using a microbalance (SDT Q600-TA Instruments Champaign, IL, USA), where 50 mg of dry sample was exposed to air-heated with temperatures of 10–700 °C with intervals of 5 °C/min, to obtain TGA combustion curve.

High-resolution transmission electron microscopy (HR-TEM) was performed by using a Philips CM-200 analytical TEM operating at 200 kV. Specimens for HR-TEM analysis were prepared by dispersing the samples in acetone via sonication for 2 min and air-drying a drop of the suspension on a perforated carbon-coated Cu° grid at room temperature (*Lara-Romero et al., 2017*).

### *Caenorhabditis elegans* killing assays

In the study, *C. elegans* Bristol N2 and CL2166 dvIs19[pAF15(Pgst-4::GFP::NLS)] worms (provided by the Caenorhabditis Genetics Center, University of Minnesota). Worms clone (CL2166) that contains the GFP reporter protein fusion into the transgene dvIs19 [(pAF15)gst-4p::GFP::NLS] III, oxidative stress-inducible GFP, whose reporter driven by the promoter of the glutathione S-transferase gene (gst-4p). In *C. elegans*, expression of gst-4 is activated by redox cycling compounds, electrophiles, and heavy metals (*Leung et al., 2013*). Worms were synchronized by hypochlorite isolation of eggs from gravid adults, and the eggs were then hatched in a S-basal medium. Specifically, L1 larvae were transferred onto nematode growth medium (NGM) (*Stiernagle, 2006*) plates seeded with *Escherichia coli* JM101 strain, which was previously grown on the plates as a food source, and incubated at 20 °C for 4–5 d until the larvae reached the young-adult phase. Worms were rinsed from the plates and washed in an S-basal medium. In each experiment, 20 worms were dispensed into each well of 36-well plates, and this was followed by incubation with 1-mL cultures media or residues obtained from sample incineration at 300 °C for 4 h. The plates were incubated at 18 °C, and live worms were scored at 12-h intervals. A worm was considered as dead when it no longer responded (moved) to a touch stimulus. Two independent assays by duplicate were conducted for each worm group.

### Microscopy

For superoxide ($O_2^{\bullet-}$) determination, worms samples were incubated with 5 μg/mL dihydroethidium (DHE; Molecular Probes, Eugene, OR, USA, Invitrogen, Carlsbad, CA, USA) at 30 °C for 30 min in the dark, washed with phosphates buffer and observed by using a fluorescent and phase contrast inverted microscope (Nikon TE300) with PlanFluor 4×, 10× dry lenses and an AmScope ML300 3.1 MP digital color camera. From the fluorescence microscopy images-obtained, the same number of nematodes from each treatment was utilized to determinate fluorescence intensity using Image J Software (NIH). Data represent the mean value ± standard error (SE) of three independent assays with *n* = 10 for each treatment. One-way ANOVA analysis was carried out, with Tukey's post-hoc test. Values for $P < 0.05$ are shown with different lower-case letters.

## RESULTS

### Fungi growth and induction of ligninolytic enzymes in cultures containing MWCNTs

Assimilation of compounds such as lignin has been widely documented and correlated with the synthesis of enzymes such as LiP, MnP, or laccase (*Baltierra-Trejo et al., 2016*; *Chen et al., 2011*; *Ting et al., 2011*). Thus, we determined the MWCNTs biodegradation capacity of *P. chrysogenum* and *P. ostreatus* fungi in cultures media supplemented with carbon nanomaterials, thereby inducing ligninolytic enzymes via the addition of lignin as a carbon source with or without glucose supplementation. Results showed that both the *P. chrysogenum* and *P. ostreatus* fungi were capable to grow on media with lignin or glucose as the sole carbon source and supplemented with MWCNTs (10 µg/mL) (Fig. 1).

Additionally, the results showed that in the *P. ostreatus* culture medium, lignin used as the carbon source induced the activities for laccase, LiP, and MnP comparing with glucose wherein it did not induce enzymatic activities at similar levels (Fig. 2). It should be noted that the activity levels of the enzymes increased when MWCNTs were supplemented with lignin in the culture medium and even when MWCNTs were used without addition of carbon source (Figs. 2A–2C). Similar results were obtained in the culture media of *P. chrysogenum* fungus (Figs. 2D–2F). Also, laccase activity exceeds LiP and MnP activities in cultures of *P. chrysogenum*, and the presence of MWCNTs increased MnP activity at similar levels than that induced with lignin (Fig. 2).

### Chemical modifications into MWCNTs by growth of *P. ostreatus* and *P. chrysogenum* fungi

Raman and infrared spectroscopy also as thermogravimetric analyses were conducted to perform an in-depth study of the effect of ligninolytic enzymes in cultures of both *P. ostreatus* and *P. chrysogenum* strains over possible chemical modifications into MWCNTs nanomaterial. The Raman spectra indicated that the addition of MWCNTs in the *P. ostreatus* cultures led to significant changes in the bands ID (~1,370 cm$^{-1}$), IG (~1,600 cm$^{-1}$), and 2D (~2,640 cm$^{-1}$) (Figs. 3A–3D). However, the presence of MWCNTs + lignin and also MWCNTs + lignin + glucose in both the *P. ostreatus* and *P. chrysogenum* cultures clearly modified the ID, IG, and 2D bands of the Raman spectra, and they were observed as broad signals with the apparition of additional bands in the range of 1,350–1,550 cm$^{-1}$ (Figs. 3B, 3C, 3F and 3G). Conversely, the addition of MWCNTs alone or with glucose in the culture medium where the *P. chrysogenum* fungus was grown did not exhibit significant changes in the ID, IG, and 2D bands of the Raman spectra thereby indicating the integral presence of MWCNTs (Figs. 3E and 3H), similar to MWCNTs without fungi exposition (Fig. 3I).

With respect to the Raman spectra, it is observed that in the MWCNTs containing samples from the fungi cultures, the IG band changed in terms of intensity and definition of peaks and exhibited a group of peaks within the range of 1,500–1,600 cm$^{-1}$ in a manner independent of the carbon source (lignin or glucose). The IG band deformation was
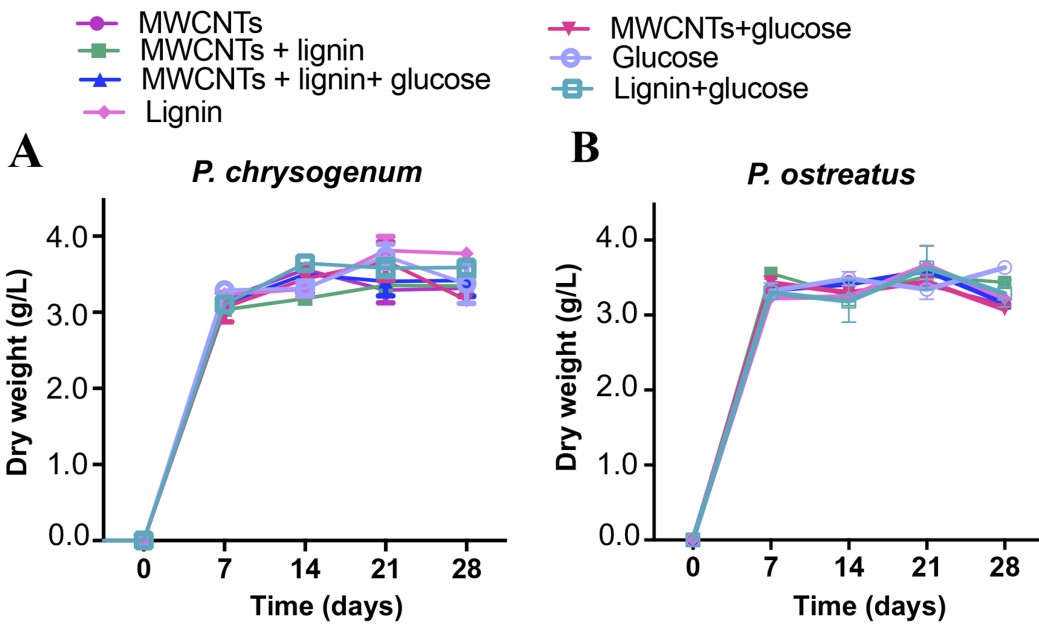

**Figure 1 Growth of *P. chrysogenum* and *P. ostreatus* on medium supplemented with MWCNTs.** Growth assays were carried out in liquid medium supplemented with MWCNTs (10 µg/mL) at 30 °C incubation by 28 days. Fungi growth is showed as biomass, determined by dry weight of culture media (g/L). (A) *P. chrysogenum*, (B) *P. ostreatus*. Results are the means of triplicate assays. SE values are indicated as bars.

significantly observed in the spectra of samples containing MWCNTs + lignin when compared with that in the glucose supplementation.

The presence of lignin in the cultures supplemented with MWCNTs and the correlation with increases in the enzymatic activities indicated the total loss of the 2D band at ~2,640 cm$^{-1}$ in the Raman spectra (Figs. 3A–3C and 3F–3G).

Additionally, the cultures of *P. ostreatus* and *P. chrysogenum* in the presence of lignin plus MWCNTs after 28 days of incubation were analyzed using FTIR spectrometry. We observed that the material obtained at beginning of the assay showed characteristic signals at ~3,260 cm$^{-1}$ and ~1,700 cm$^{-1}$ that correspond to O–H functional groups (Fig. 4, black and light blue lines). These radicals are characteristic of disordered regions of MWCNTs, which conforms the nucleation sites for hydrogen atoms. Other signals attributed to stretch vibrations of C-H at ~2,200 cm$^{-1}$ and C–H$_X$ groups of the disordered sp3 region and vibration rings at 600 cm$^{-1}$ were characteristics of MWCNTs (*Al-Rekaby, 2018*; *Lehman et al., 2011*). Conversely, we observed that the FTIR spectra of the culture residue from *P. ostreatus* cultivated with MWCNTs + lignin indicated the absence of the characteristic bands of the MWCNTs, also a characteristic peak of the COOH functional radicals was identified at ~1,700 cm$^{-1}$ (Fig. 4). The FTIR spectrum of the culture residue from *P. chrysogenum* grown on medium with MWCNTs + lignin indicated the lower intensity of the characteristic peak of the O–H groups at ~3,260 cm$^{-1}$ and absence of peaks characteristic of MWCNTs. Contrariwise, the presence of COOH signal at ~1,700 cm$^{-1}$ and CO groups at ~1,100 cm$^{-1}$ were identified (Fig. 4).

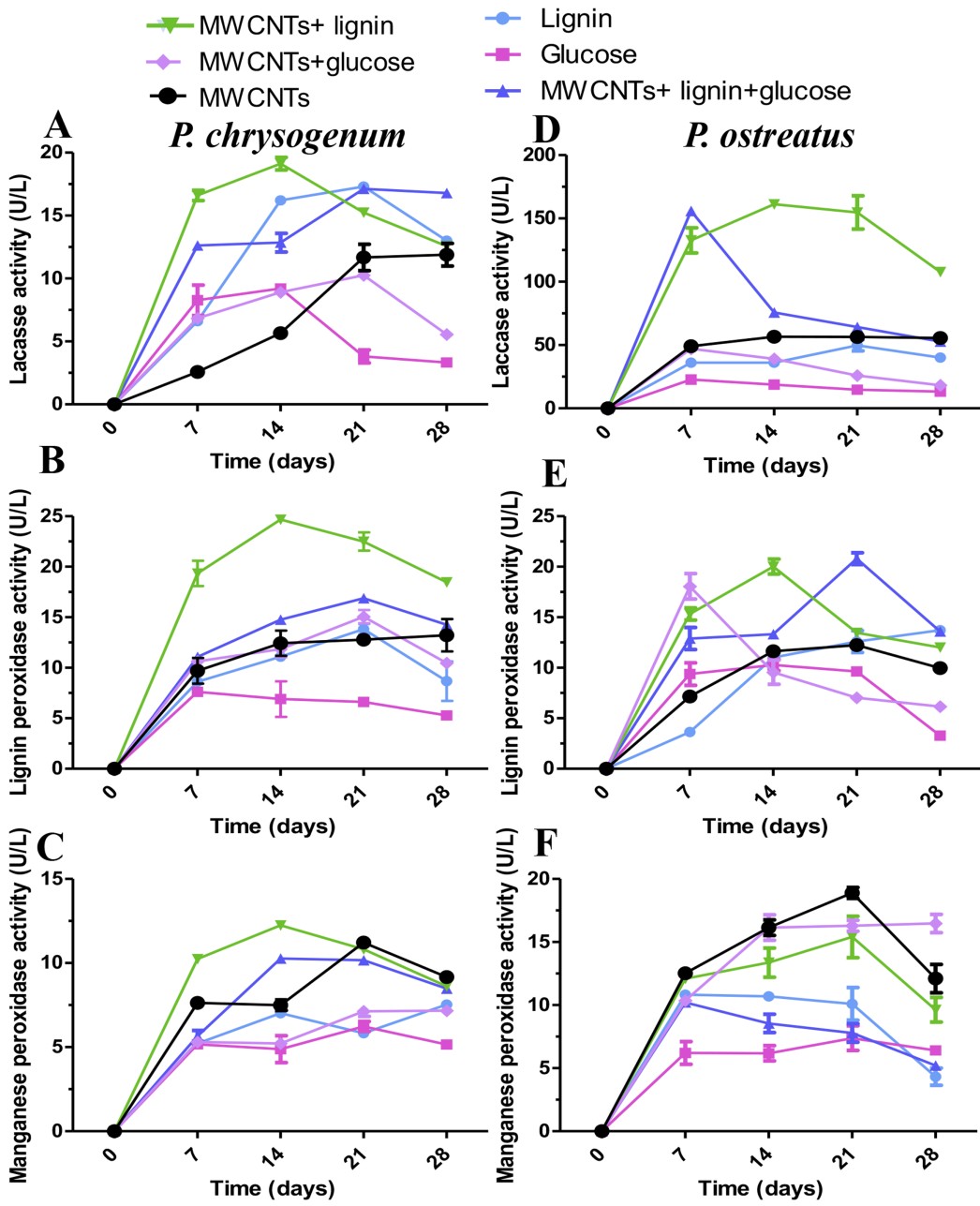

**Figure 2 Ligninolytic activities on cultures of *P. chrysogenum* and *P. ostreatus* grown in presence of MWCNTs.** Fungi growth was carried out in liquid medium supplemented with MWCNTs (10 µg/mL) at 30 °C incubation by 28 days. Effect of carbon sources on ligninolytic activities at 28-day of culture of *P. chrysogenum* (A–C) and *P. ostreatus* (D–F) fungi. Enzyme activity units are determined by dry weight from 100 mL of culture media. Results are the means of three independent assays. SE values are indicated as bars.

## Structural modifications into the MWCNTs by growth of *P. ostreatus* and *P. chrysogenum* fungi

Chemical changes in the MWCNTs observed in Raman and IR spectroscopy analyses were examined by high resolution transmission electronic microscopy (HR-TEM). It was

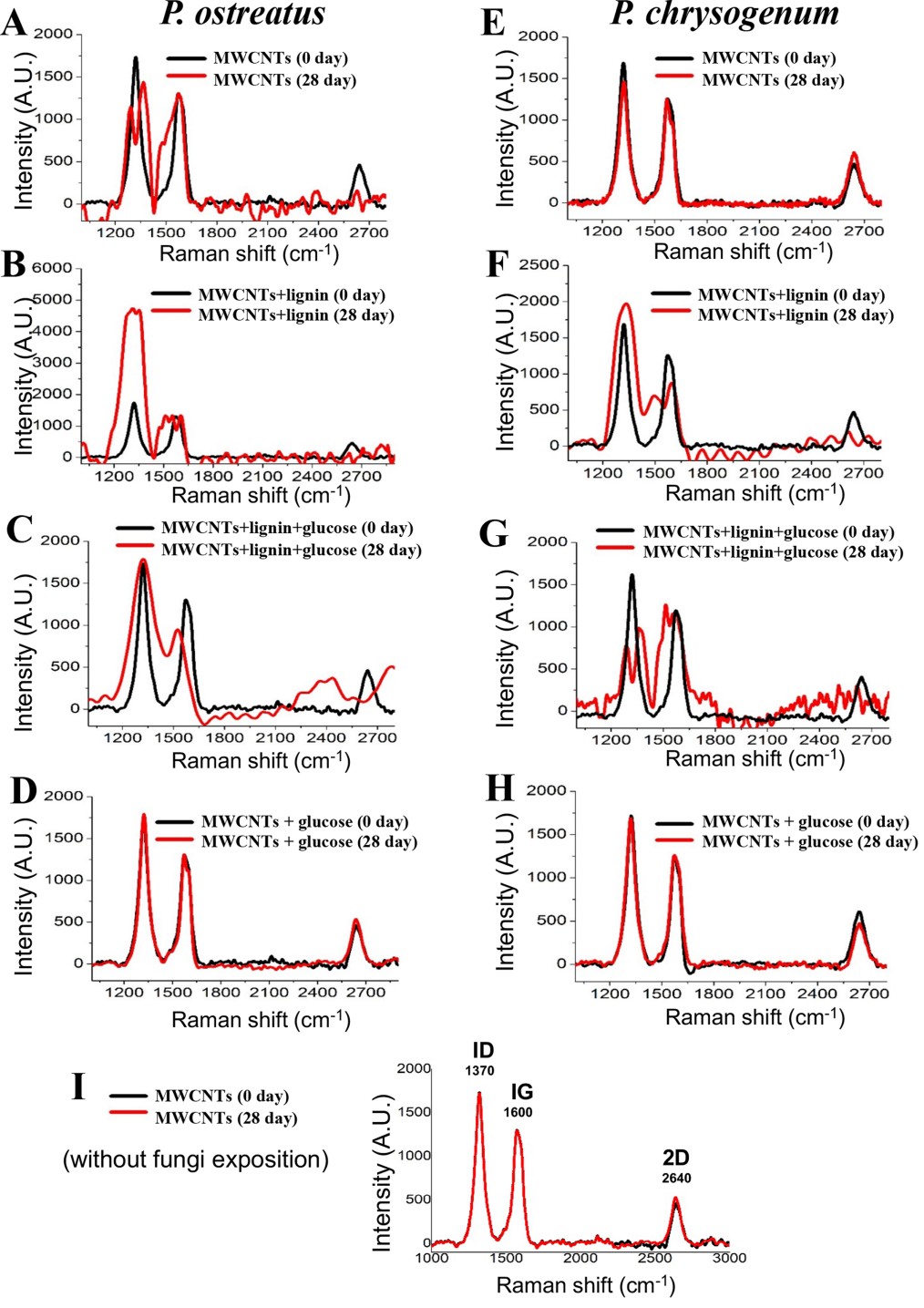

**Figure 3 Chemical modifications into the MWCNTs by *P. ostreatus* and *P. chrysogenum* growth determined by Raman spectroscopy.** Samples of fugal cultures supplemented with MWCNTs were dried and incinerated prior to Raman analysis taken from culture of *P. ostreatus* (A–D) and *P. chrysogenum* (E–H) fungi. (I) MWCNTs control samples without fungi exposition. Representative Raman scattering spectra (He-Ne laser emitting at 514 nm) of the culture samples are shown; MWCNTs at the beginning of the assay (0 day, black lines) and cultures samples after 28 days of incubation (red lines). Characteristic bands of MWCNTs are shown: ID band (~1,370 cm$^{-1}$), IG band (~1,600 cm$^{-1}$), and 2D (G′) band (~2,640 cm$^{-1}$).
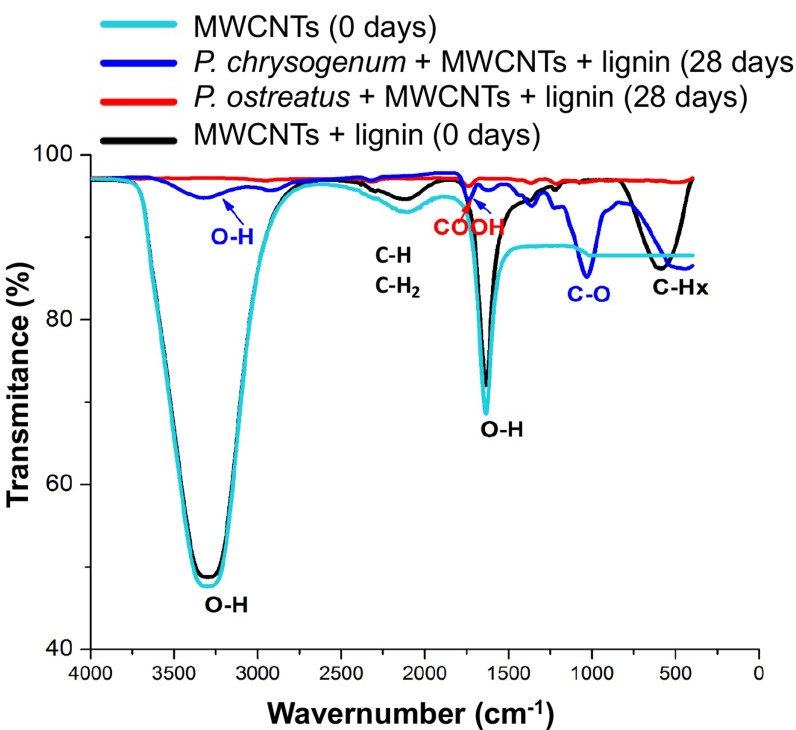

**Figure 4 Chemical modifications into the MWCNTs by *P. ostreatus* and *P. chrysogenum* growth determined by FTIR spectroscopy.** Samples of fugal cultures supplemented with MWCNTs were dried and incinerated prior to FTIR analysis. Cultures with MWCNTs grown with the *P. ostreatus* (red line) and *P. chrysogenum* (blue line) analyzed after 28 days of incubation. Controls are shown (black and light blue lines). Signals of radicals or functional groups are indicated. Results were conducted by triplicate and representative graph is shown.

observed that the structure of the MWCNTs was heterogeneous after the growth of *P. chrysogenum* and *P. ostreatus* fungi in media supplemented with MWCNTs. The HR-TEM images showed a clear decrease in the amount of MWCNTs aggregates (Figs. 5A–5J). The intensity of structural changes in the MWCNTs were dependent on the fungus strain and the treatment that was applied. The structural modifications were more evident in the cultures of *P. ostreatus* using MWCNTs + lignin as carbon sources (Figs. 5A–5E). It should be noted that structural changes in the surface of the nanotubes were observed and included loss of layers and changes in both the internal and external diameters of the MWCNTs structures (Figs. 5N and 5O). The average internal and external diameters of the synthetic MWCNTs that exhibit homogeneous structures were around 4 nm and 20 nm, respectively (Figs. 5K–5M); while the diameters of MWCNTs from the samples analyzed after the cultivation for 28 days of the *P. ostreatus* fungus in the medium supplemented with MWCNTs + lignin were approximately 8 nm and 16 nm, respectively (Figs. 5N and 5O).

In addition, thermogravimetric analysis (TGA) of dried samples from the cultures of the *P. ostreatus* and *P. chrysogenum* fungi showed that the MWCNTs amount clearly decreased after 28 days of growth in the medium supplemented with lignin or glucose (Fig. 6). Compounds that are burned below 300–480 °C correspond to lignin/glucose and

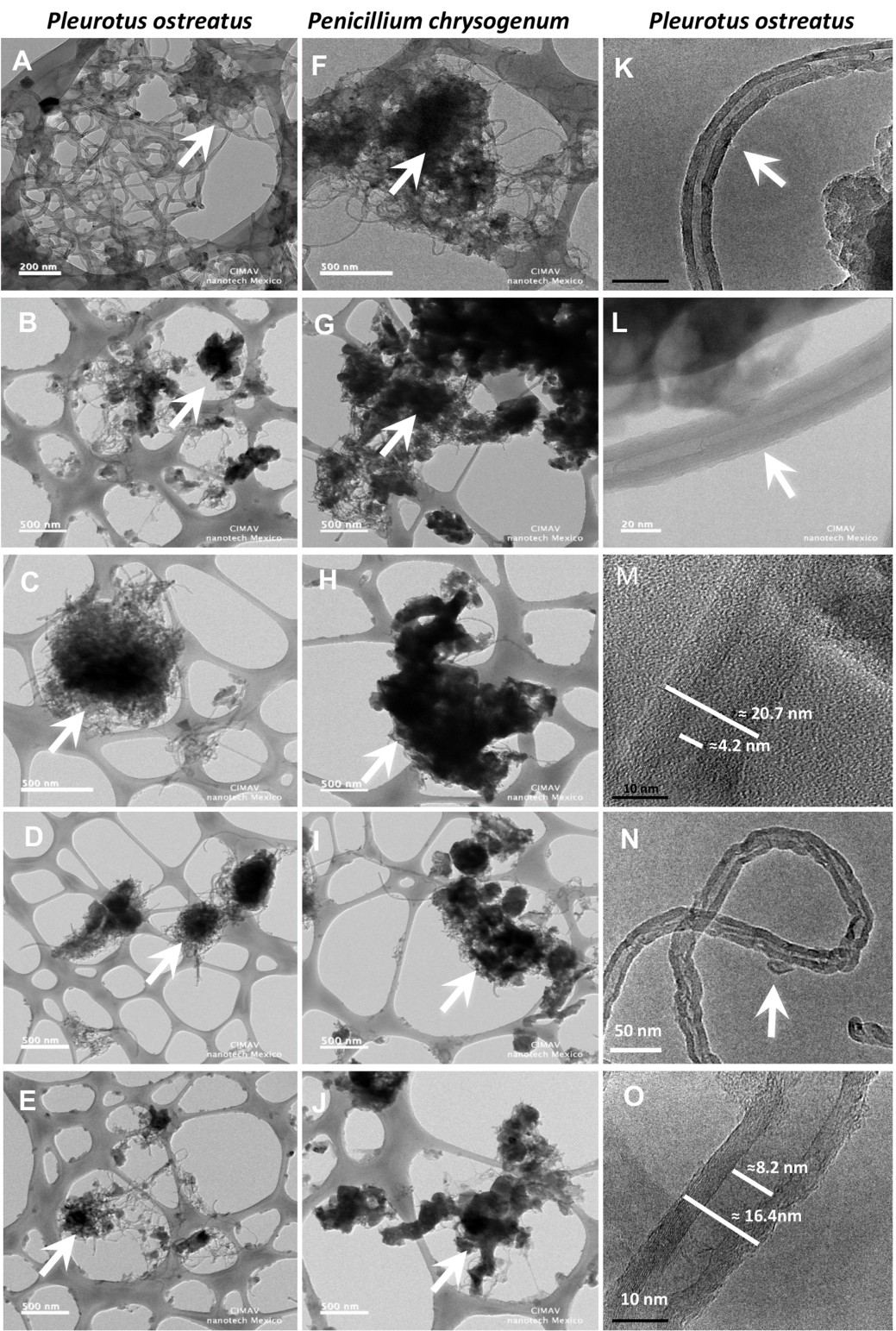

*Plerotus ostreatus*     *Penicillium chrysogenum*     *Pleurotus ostreatus*

**Figure 5 Structural modifications into the MWCNTs by *P. ostreatus* and *P. chrysogenum* growth observed by HR-TEM.** Samples of fugal cultures supplemented with MWCNTs were dried, incinerated and mounted on perforated carbon-coated Cu^o grid to observed by transmission electronic microscopy. *P. ostreatus* fungus cultures incubated with (A) MWCNTs for 0 days, and after 28 days of

**Figure 5 (continued)**
incubation (B and C) MWCNTs + glucose at 28 day, (D) MWCNTs + lignin at 28 day, and (E) MWCNTs lignin + glucose at 28 day. *P. chrysogenum* fungus cultures incubated with (F) MWCNTs for 0 days, and after 28 days of incubation with MWCNTs (G and H) MWCNTs + glucose at 28 day, (I) MWCNTs + lignin at 28 day, and (J) MWCNTs + lignin + glucose at 28 day. Increased-amplification HR-TEM images of *P. ostreatus* fungus cultures incubated with MWCNTs + lignin at 0 day (K–M) and cultured with MWCNTs + lignin at 28 day (N and O). MWCNTs aggregates and nanotube structures are indicated with arrows.                                

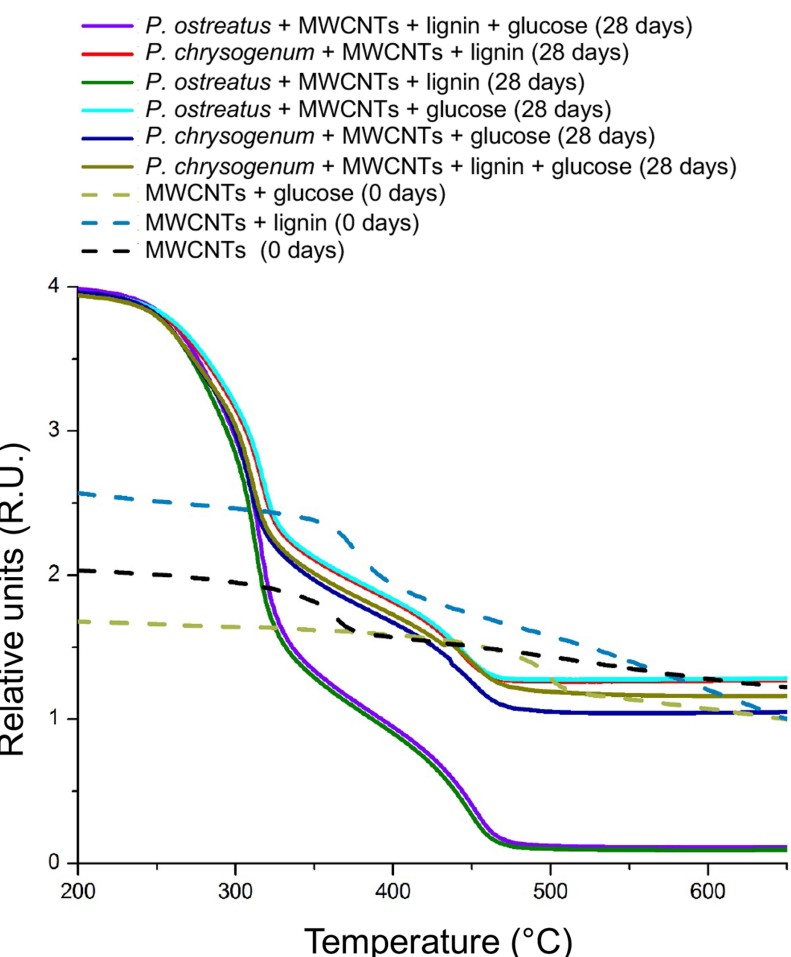

**Figure 6 Mass loss on cultures of *P. ostreatus* and *P. chrysogenum* grown with MWCNTs by thermogravimetric analyses (TGA).** Samples of fugal cultures supplemented with MWCNTs were dried prior to TGA analysis. Before incubation (0 days) and after 28 days. *P. ostreatus* and *P. chrysogenum* cultures added with MWCNTs were analyzed after 28 days of cultivation. Representative plots are shown.                                

cellular biomass; while that compounds that remains unburned above >500–600 °C suggest to be crystalline carbon nanomaterials such as MWCNTs and graphene. In the TGA analysis, the MWCNTs amount in the ~1.0 mg of each samples at the beginning of the experiment was submitted to incineration around 500–700 °C, the determination of sample weight were reduced in the treatments with *P. ostreatus* after 28 days of

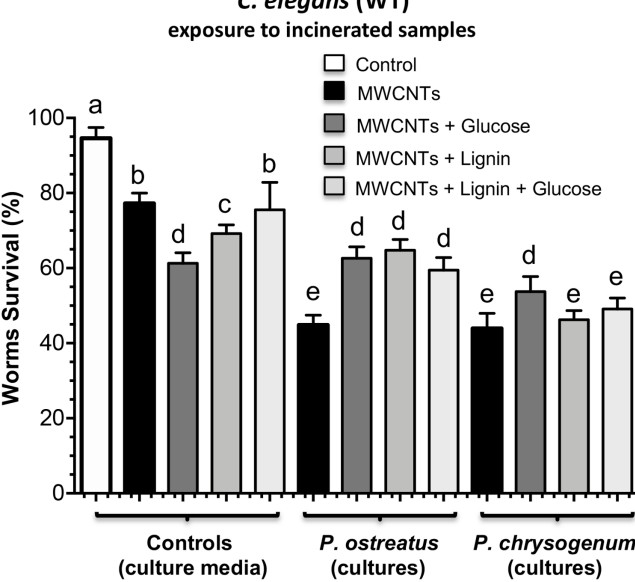

**Figure 7** Toxic effects on *C. elegans* worms exposed to cultures of *P. ostreatus* and *P. chrysogenum* grown with MWCNTs. Samples from media supplemented with MWCNTs and from fugal cultures grown by 28 days were dried and incinerated prior to worms survival test. Adult *C. elegans* worms (20 individuals) were incubated in 1 mL of culture media (control) or residues obtained after incineration at 300 °C of samples from the *P. ostreatus* and *P. chrysogenum* fungi grown on medium with MWCNTs. Worms survival was determined at 72 h of exposure as described in Materials and Methods. Conditions of treatment are indicated. Bars represent the mean value ± standard error (SE) of three independent assays with *n* = 20 for each treatment. One-way ANOVA analysis was carried out, with Bonferroni post-hoc test. Values for $P < 0.05$ are shown with different lower-case letters. Kinetics of worm survival are shown in Fig. S2.

incubation. Interestingly, in the treatment with *P. ostreatus* grown on lignin + glucose + MWCTNs, the major weight loss in sample was observed, suggesting that *P. ostreatus* degraded MWCNTs significantly more efficient than *P. chrysogenum* (Fig. 6).

## Toxic effects of MWCNTs in *C. elegans* worms

With respect to the possible toxic effects of the MWCNTs over the live organisms, we determined their toxicity by using supernatants from cultures of fungi containing MWCNTs on the toxicological animal model of *C. elegans* wild type strain and the recombinant worms clone (CL2166) that contains the GFP reporter protein fusion, oxidative stress-inducible (*Leung et al., 2013*).

Worms were placed in the presence of cultures media filtered and culture media incinerated at 300 °C and worm survival was determined over time as described in Material and Methods. The culture medium containing MWCNTs exhibited ~80% survival of nematodes versus ~95% with medium alone at 72 h of worms exposure; interestingly, the addition of glucose, lignin, or both in the medium strongly increased the culture medium toxicity, showing ~70–80% worms survival (Fig. 7). On the other hand, the samples from cultures of both *P. ostreatus* and *P. chrysogenum* that were submitted to incineration to avoid effect of biological-origin compounds present in the culture media after fungi growth, exhibited different levels of survival on the nematodes, corresponding

between 45% and 60% when they were incubated by 72 h with a medium containing MWCNTs + glucose, MWCNTs + lignin, MWCNTs + lignin + glucose, or MWCNTs alone (Fig. 7; Fig. S2 in Supplemental Material). Interestingly, in the samples from both fungi cultures grown on MWCNTs + glucose, MWCNTs + lignin, MWCNTs + lignin + glucose, worms survival significantly increased, comparing to that containing MWCNTs alone, being a clearer effect with the *P. ostreatus* fungus.

To deepen in the MWCNTs mechanism of toxicity involved, we determined the generation of reactive oxygen species (ROS) in the WT nematode strain. The results showed that the *C. elegans* WT increased the fluorescence dependent of anion superoxide amounts under the MWCNTs treatment (Figs. 8A–8D). This effect was further confirmed when the worm clone (CL2166) GFP oxidative stress-inducible was used, the fluorescence by GFP activity was significantly increased and observed in all the body of nematodes treated with MWCNTs (Figs. 8E–8P).

In addition, we observed that the CL2166 worm clone showed a basal fluorescence when it was exposed to fungal culture medium, also like adding lignin (Figs. 8E–8G). However, when MWCTNs, MWCNTs + glucose, or MWCNTs + glucose + lignin were incinerated and added to the worm suspensions, their fluorescence significantly was increased (Figs. 8H–8J). Importantly, the intensity of fluorescence (ROS generation) in the worms was increased when used incinerated fungi culture medium from *P. ostreatus* and *P. chrysogenum* (grown for 28 days on a medium with MWCNTs), indicating increases of toxicity (ROS increase) on worms (Figs. 8K–8P). Quantitative determination of GFP-fluorescence in the worms corroborated the increase of ROS induction in the worms treated with the incinerated culture medium from both fungi (Fig. 9).

## DISCUSSION

Nanotechnology is an emerging area of multidisciplinary interest because the extraordinary physical and chemical properties of nanoparticles (*Zaytseva, 2016*). The MWCNTs demand is in growing; however, its accumulation in the environment due to their high structural stability, represents great environment risks to consider, since MWCNTs have been associated with toxic effects in several biological systems (*Ge et al., 2016*; *Shin, Song & Um, 2015*). An alternative to biodegrade MWCNTs or reduce collateral environmental risks of these nanoparticles is the use of mycorremediation with ligninolytic fungi such as *P. ostreatus* and *P. chrysogenum*, which by synthesizing extracellular enzymes such as laccase, lignin-peroxidase (LiP), and manganese-peroxidase (MnP) represent an alternative to mitigate the ecological damage (*Fujii et al., 2013*; *Janusz et al., 2013*).

The use of white rot fungi to restore environments contaminated with highly recalcitrant organic xenobiotics have been widely documented, due to the ability of basidiomyces genus to synthesize high amounts of peroxidases and laccases (*Fujii et al., 2013*). *P. ostreatus* belongs to the Basidiomycetes group, which are ligninolytic fungi considered among the most efficient microorganisms to mineralize lignin. However, little has been documented on the enzymatic activity in deuteromycetes such as *P. chrysogenum*. In this work, we were able to demonstrate that lignin stimulates the activity of LiP and

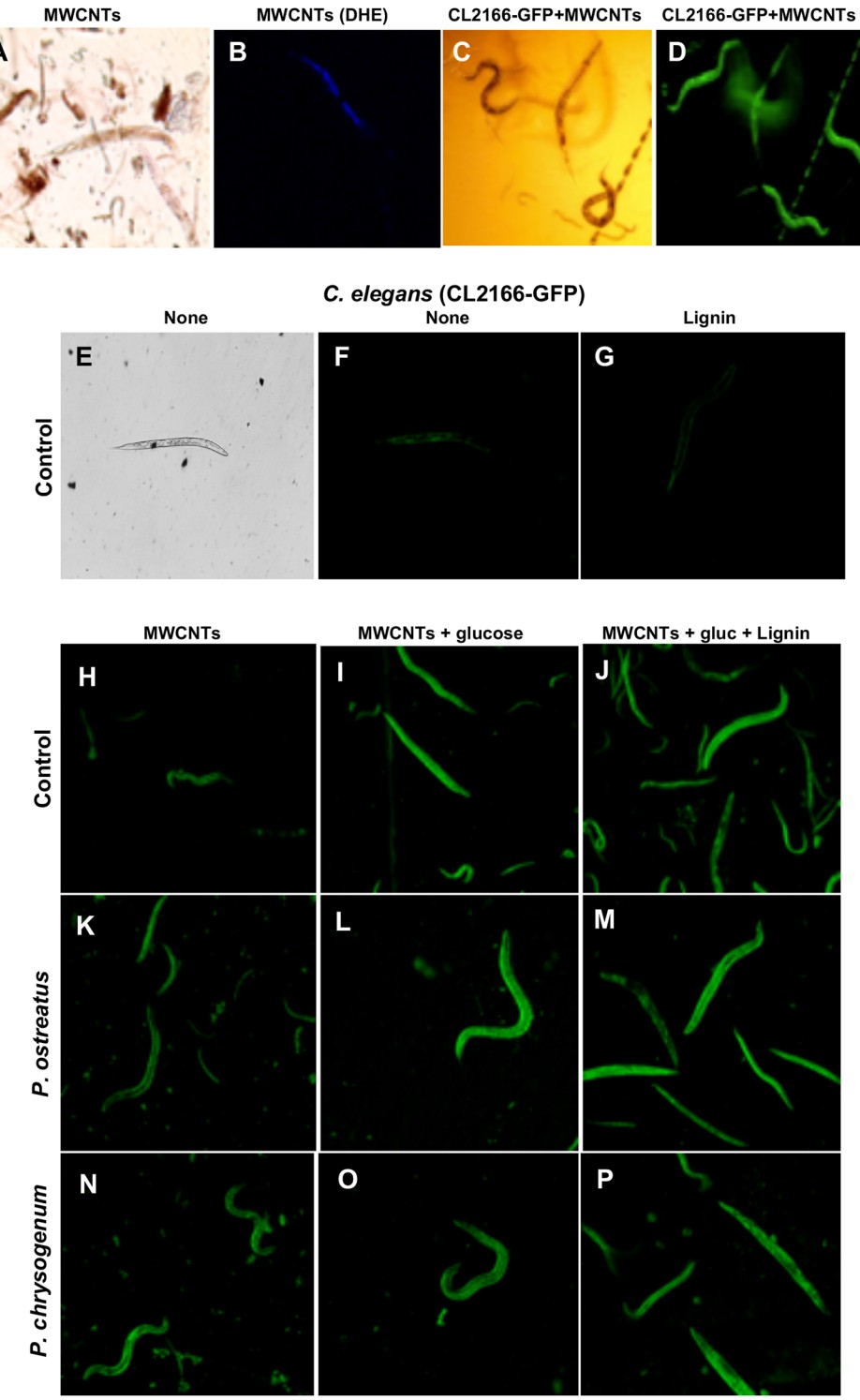

**Figure 8 Microscopy fluorescence images of *C. elegans* worms treated with MWCNTs.** (A–D) *C. elegans* WT treated with 10 μg/mL MWCNTs and superoxide generation was developed by using the Dihydroethidium fluorescent probe (DHE). (A) Worms observed with clear field at 10× magnification. (B) Worms treated with MWCNTs by 4 h exposition and using DHE as probe, after observed by fluorescent field at 10× magnification. *C. elegans* (CL2166) GFP oxidative stress-inducible clone treated with 10 μg/mL MWCNTs; (C) clear field image and (D) fluorescent field at 10× magnification.

**Figure 8 (continued)**
(E–G) *C. elegans* CL2166-GFP treated without or with lignin as indicated. (H–P) *C. elegans* CL2166-GFP worms treated with samples taken from culture media of the *P. ostreatus* and *P. chrysogenum* fungi grown on medium with 10 µg/mL MWCNTs as indicated. Samples of fugal cultures were dried and incinerated prior to test. Worms observed by fluorescent field at 10× magnification. Representative images are shown.                                                                                                  

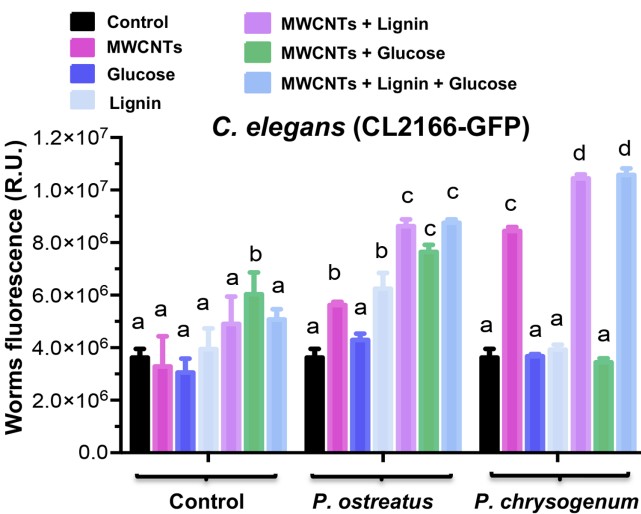

**Figure 9 Reactive oxygen species production determined by fluorescence on the *C. elegans* CL2166-GFP worms treated with MWCNTs.** Worms treated by 24 h with the compounds at concentrations used in the culture media are indicated as control. Worms exposed to incinerated samples from fungi cultures grown on medium with MWCNTs are indicated as from *P. ostreatus* and *P. chrysogenum*. Fluorescence was determined using the Image J software in at least 10 worms of each treatment as observed in imagens showed in Fig. 8C. Bars represent the mean value ± standard error (SE), $n = 10$ for each treatment. One-way ANOVA analysis was carried out, with Tukey's post-hoc test. Values for $P < 0.05$ are shown with different lower-case letters.                                                         

MnP in the *P. chrysogenum* cultures. In addition, it was observed that the presence of MWCNTs also influenced the stimulation of peroxidases activity in both the *P. chrysogenum* and *P. ostreatus* cultures (Figs. 2A–2C). It should be noted that the presence of MWCNTs or when supplemented with lignin triggered the activity of ligninolytic enzymes was triggered. The aforementioned result was in agreement with that reported in *T. versicolor* and *P. tremellosa* isolates wherein the presence of SWCNTs positively modified its lignocellulolytic activity (*Berry, Filley & Blanchette, 2014*). Relevantly, our findings indicate that both fungi species were capable to grow efficiently on medium supplemented with MWCNTs, demonstrating that this nanomaterial was no toxic at least to the concentration used (10 µg/mL of MWCNTs) and an efficient fungal growth and ligninolytic enzymes excretion was observed. The results confirm that lignin played a key role in the induction of ligninolytic enzymes such as those reported in basidiomycetes during the lignin depolymerization process (*Baltierra-Trejo et al., 2016*; *Cañas & Camarero, 2010*; *Nousiainen et al., 2014*).

Moreover, the changes in Raman bands indicated the occurrence of structural modifications in the surface of MWCNTs (Fig. 3). The difference in the band IG band

confirm de increment of defects in the crystalline structure of MWCNTs in the treatments with MWCNTs lignin and MWCNTs + lignin + glucose with both fungi, where the increase in height and width in this band, corresponds to the structural disorder in the graphitized materials (*Liu et al., 2014*). The changes in spectrum corresponding to the sp3 bonds of the distortions of the network in the curved graphene sheets or the ends of the carbon nanotubes (*Lehman et al., 2011*), and loss of crystallinity by oxidation of these nanoparticles, it can be attributed to ligninolytic enzymes, wherein activities clearly increased in the cultures during the assimilation of lignin. Thus, the changes in the IR spectra suggest that both the *P. ostreatus* and *P. chrysogenum* fungi modified the structural composition of MWCNTs. The presence of new functional groups, such as COOH or CO as described (*Lehman et al., 2011*), confirms that the excretion/production of ligninolytic enzymes provoked chemical modifications in the composition of the MWCNTs. Furthermore, the incorporated functional groups in the MWCNTs suggested that a biotransformation process occurred. Our results indicate that both fungi have the ability to degrade/mineralize the MWCNTs in a dependent manner on the available carbon source. Thus, decreases in the amount of MWCNTs observed in the HR-TEM sample observations and TGA were correlated with the Raman and IR spectroscopy analyses; thereby indicating that the MWCNTs sustained chemical and structural modifications during the fungi growth on culture media supplemented mainly with lignin. This indicated the occurrence of biotransformation and biodegradation/mineralization processes associated with the ligninolytic fungal enzymatic activities. As revealed by *Liu et al. (2014)*, the intensity of the 2D band is correlated with the number of graphene layers that constitute the structure of MWCNTs, thereby suggesting a reduction in the thickness of the MWCNTs following the growth of *P. chrysogenum* and *P. ostreatus* fungi on the cultures containing MWCNTs + lignin, suggesting that the role of lignin is the induction of peroxidases, which chemically modify the chemical composition of the MWCNTs material.

In addition, the results obtained with both fungi indicate that the structural transformations of MWCNTs induced by the fungal enzymes also in presence of lignin or glucose increase the toxic effects than when nonculture the fungi. It should be noted that diverse effects on the nematodes as our in vivo model are caused by the dried samples from the fungi cultures that contain MWCNTs. First, the MWCNTs can be modified by the fungus strains when used with additional carbon source, without apparent toxic effects over their growth. However, MWCNTs exhibits a certain level of toxicity for the nematodes (~25% death at 72 h exposure). Second, the presence of other carbon sources such as lignin or glucose which induce ligninolytic enzymes increases its toxicity over the nematode model (~50–70% death). This effect is potentially explained due to the chemical and structural modifications in the MWCNTs observed, such as have been described to occur in microbial communities (*Ge et al., 2016*; *Wu et al., 2019*). However, we can no to explain why the medium supplemented with MWCNTs + glucose was more toxic to the worms than MWCNTs alone, but after fungi growth is effect was reverted as expected (Fig. 7).

These chemical and structural modifications are clearly demonstrated in the Raman, IR, and HR-TEM analyses, which were in correlation with the results of the biological effects such as growth and peroxidases production. In agree with TGA results, *P. ostreatus* degraded MWCNTs more efficient than *P. chrysogenum* fungi, indicating that almost no residues of MWCNTs present on cultures from *P. ostreatus* induced lower mortality in *C. elegans* worms compared to the respective MWCNT amounts from *P. chrysogenum* samples (Fig. 7). The toxic effect observed by the MWCNTs addition against the *C. elegans* nematode further confirms an exacerbated increment of ROS generation as a toxicity mechanism (Figs. 8 and 9). This effect could be due to the structural modification by the COOH, CO, or OH radicals identified on the MWCNTs, which functional groups which have been widely described to increase the toxicity in nanomaterials (*Magrez et al., 2006*; *Zhou et al., 2017*).

In the study, the ability of fungi species such as *P. ostreatus* and *P. chrysogenum* to grow on a culture medium with the presence of highly recalcitrant nanoparticles (MWCNTs) was evidenced, which toxic effects of the MWCNTs over the fungi were not observed. Our findings suggest that the degree of toxicity of the synthetic MWCNTs in the in vivo nematode model was modified by the fungal laccase and peroxidases activity excreted to the culture media, and these correlated with chemical modifications observed. The toxic effect of the MWCNTs against the *C. elegans* nematode involved an increased ROS generation. Interestingly, the results obtained in the study indicate the potential ability of the fungi species *P. ostreatus* and *P. chrysogenum* to produce structural biotransformation on MWCNTs in a relatively short time, indicating that although the toxicity was increased under certain growth conditions, the amounts of the carbon nanomaterial was significantly decreased, as confirmed by spectroscopy analyses. The fact is mainly attributed to the ability of the fungi to synthesize ligninolytic enzymes, wherein extracellular enzymes are related with the assimilation mechanism of residual lignin and other persistent or recalcitrant compounds. The findings indicated that the ligninolytic enzymes induced by lignin are key in the chemical and structural modifications observed in the MWCNTs, being these modifications important in the toxicity degree and environmental persistence of these carbon nanomaterials.

## CONCLUSIONS

The results obtained in the work further confirmed that the MWCNTs can be subject to biotransformation events in the environment via the mycorremediation, thereby modify its toxic effects over others organisms. This study contributes with a better understanding of the interactions between engineered carbon nanomaterials and microbial communities, as their beneficial as toxicological effects over the ecosystem.

## ACKNOWLEDGEMENTS

*Caenorhabditis elegans* strains were provided to J. Altamirano and A. Guzmán by the Caenorhabditis Genetics Center (University of Minnesota, Minneapolis, MN, USA).

### Funding

This research was funded by CONACYT (256119) and C.I.C. 2.14/UMSNH grants. Gladys Juárez-Cisneros received a scholar-ship of CONACYT. The funders had no role in study design, data collection and analysis, decision to publish, or preparation of the manuscript.

### Grant Disclosures

The following grant information was disclosed by the authors:
CONACYT: 256119 and C.I.C. 2.14/UMSNH.

### Competing Interests

Jesús Campos-García is an Academic Editor for PeerJ.

### Author Contributions

- Gladys Juárez-Cisneros performed the experiments, analyzed the data, prepared figures and/or tables, authored or reviewed drafts of the paper, and approved the final draft.
- Jesús Campos-García conceived and designed the experiments, analyzed the data, prepared figures and/or tables, authored or reviewed drafts of the paper, and approved the final draft.
- Sharel Pamela Díaz-Pérez performed the experiments, authored or reviewed drafts of the paper, and approved the final draft.
- Javier Lara-Romero performed the experiments, authored or reviewed drafts of the paper, and approved the final draft.
- Dhirendra Kumar Tiwari performed the experiments, analyzed the data, authored or reviewed drafts of the paper, and approved the final draft.
- Juan Manuel Sánchez-Yáñez performed the experiments, authored or reviewed drafts of the paper, and approved the final draft.
- Homero Reyes-De la Cruz analyzed the data, authored or reviewed drafts of the paper, and approved the final draft.
- Sergio Jiménez-Sandoval analyzed the data, authored or reviewed drafts of the paper, and approved the final draft.
- Javier Villegas conceived and designed the experiments, analyzed the data, prepared figures and/or tables, authored or reviewed drafts of the paper, and approved the final draft.

### Data Availability

  Raw data are available as Supplemental Files.

### Supplemental Information

Supplemental information for this article can be found online at http://dx.doi.org/10.7717/peerj.11127#supplemental-information.

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
