# Peer review of "Ligninolytic activity of the Penicillium chrysogenum and Pleurotus ostreatus fungi involved in the biotransformation of synthetic multi-walled carbon nanotubes modify its toxicity"

_PeerJ, doi:10.7717/peerj.11127_

## Round 0.1 · original submission · Major Revisions

When revising the manuscript, please pay special attention to improving the clarity of experimental descriptions and figure captions as the Reviewers have pointed out. Importantly, check for any mistakes in figures (for example, Fig. 2A and B seem to be the same as also mentioned by the Reviewers). Please consider modifying the presentation of the data for clarity, e.g., showing the data in Fig. 7 in multiple separate panels. Reviewers have also indicated missing (control) data in some of the figures which would be important to add for the correct interpretation of the study results.

Reviewer 1 ·

Basic reporting

no comment

Experimental design

no comment

Validity of the findings

no comment

Additional comments

The topic of biotransformaton of MWCN by fungi is very crucial for the environmental helath. This study
investigated the ability of fungi to MWCNTs biotransformation by peroxidases induction and
evaluation in vivo of its toxicity using Caenorhabditis elegans as a model. The manuscript was overall well organized and easy to follow. Some comments listed below may be helpful to improve the manuscript.
1. Fig. 1 cannot clearly show the growth trend. It is recommended to set a breakpoint on the ordinate.
2. Fig. 2A and Fig. 2B are almost identical, please ensure the raw data.
3. There are many kinds of fungi, why choose Pleurotus ostreatus and Penicillum chrysogenum as the research objects.
4. L109 Need to supplement the culture deposit number of GenBank.
5. L136 The ɛ of laccase is wrong. Please re-calculate laccase activity.
6. Please elaborate the significance of the research in the nature environment in "Results and Discussion".
7. I suggest anthors to improve the sharpness of Figure. 3.

Reviewer 2 ·

Basic reporting

The article is written in clear, professional English.
The text of the article is supported by a sufficient number of references to modern scientific publications.
The structure of the article meets the existing requirements, the figures are relevant, of sufficient resolution.
In order for the results to confirm the hypotheses put forward by the authors, additional controls are required. Questions on setting controls have been added to the "Validity of the Findings" section.

Experimental design

The article corresponds to the Biological Sciences.
Research question well defined, relevant & meaningful. It is stated how research fills an identified knowledge gap.
The technical level of experimental work is sufficient to achieve these goals.
It is necessary to describe more clearly the methods for separating MWCNTs from the cells of fungal cultures; it is not clear what was analyzed - pure MWCNTs without organic matter or MWCNTs with cells. How did you get rid of organic matter?

Validity of the findings

1.Fig. 1. The dry weight is given in grams, for what volume? It is better to give the biomass concentration in g / l.
2. Fig. 2. Ligninolytic activities on P. chrysogenum and P. ostreatus cultures grown in presence of MWCNTs: The black line on the graph corresponds to the MWCNTs. In this case, what was the carbon source for the fungi, was it a yeast extract? When glucose and lignin were added, was it yeast extract + glucose, etc.?
3. Fig. 3. Raman spectra of P. ostreatus and P. chrysogenum cultures grown with MWCNT supplementation: The figure shows the Raman spectrum of MWCNTs modified under the influence of fungal cultures? In this case, adjustments should be made to the caption to the figure; from the caption, one might think that this is a spectrum of fungal cells. I believe that control of MWCNT should be presented in these media, but without fungal cultures, 0 and 28 days.
4. Fig. 4. How was MWCNT separated from lignin? Precisely the spectra of MWCNTs are presented, without lignin impurity? Where is the spectrum of pure MWCNTs without lignin?
5. Fig. 5. What is shown in the figure K-M and H-O - these are nanotubes? It should be pointed out, it is not clear from the figure. The figure caption indicates that these are images of fungal cultures. In the rest of the drawings, is it the backing grid or cells? It should be pointed out. The images must be signed, at first glance they are incomprehensible, only nanotube aggregates are clearly identified.
6. Fig. 6. Data missing: 0 day for cultures + MWCNT + glucose, cultures + MWCNT + glucose + lignin. Or it is incomprehensibly described in the figure captions. How was point 0 taken - is it a culture medium containing nutrients (yeast extract?) and an inoculum, or pure MWCNT + lignin?
7. Fig. 9. There is no data on how the culture medium of pure fungal cultures affects the worms (without the addition of MWCNT).
8. Fig. 10. It should be clearer to decipher a, b, c, d in the figure (or more clearly describe the statistical analysis in the Materials and Methods). How were the units of fluorescence correlated with the number of living worms? Was the increase in fluorescence due to more live worms, or due to oxidative stress?
9. How to explain that MWCNT + glucose gives an increased fluorescence of recombinant worms (Fig. 10), while glucose in the fungal cultivation medium suppresses the activity of the studied enzymes (Fig. 2)? According to the authors, these enzymes modifying MWCNTs and increasing their toxicity (line 380-383) which leads to oxidative stress and increased fluorescence of the worms. It is worth discussing these results in more detail.

Additional comments

I believe that the article should clearly indicate whether the modification of MWCNTs was studied without admixture of cells and organic matter, in this case, describe in Materials and methods how the MWCNTs were separated from cells that could partially be in the culture medium, forming aggregates with MWCNTs. Then it is necessary to correct the captions to the figures, to avoid phrases like "Raman spectra of P. ostreatus and P. chrysogenum cultures ..."

·

Basic reporting

Overall, the English language is OK.
The introduction must contain the remarks of "mycoremediation." Mushroom is used for the one of the main strain for that.

Experimental design

The manuscript meets the aims and scope of the journal.
Some data for the conclusion are missing; mass balance for the degradation (GC data needed).
It is unclear why the growth curves are the same between glucose and lignin as well as other experimental sets (Fig. 1).
For TGA analyses, the physical mix of mycelium and MWCNT must be included in the experimental sets for comparing to 28 day data.
Fig. 7 is too complicated; it should be rearranged for clear results.

Validity of the findings

The data for fungal growth and enzyme production are ambiguous. The authors should explain how the three ligninolytic enzymes attack the lignin and CNT based on the enzyme production.
In Fig. 2, A and B are the same. The growth patterns are the same btw the two strains, but the enzyme production patterns are rather different. This phenomena should be comprehensively discussed with appropriate references.
The mass balance are missing and the metabolite data should be included for the conclusion.

Additional comments

The topic is interesting and suitable for the journal. However, it cannot be published unless some data and comprehensive discussion are to be supported.

---

## Round 0.2 · Minor Revisions

Thank you for submitting the revised manuscript and addressing the Reviewers’ comments. While the manuscript has been improved and the Reviewers had no additional questions, there are still a few aspects which seem important to be addressed and elaborated on.

First, the results presented in Figure 6 appear to be misinterpreted in the Results and Discussion (lines 342-345). The TGA results presented in Figure 6 implicate that the two treatments which resulted in almost complete loss of MWCNTs were both P. ostreatus treatments, purple and dark green line representing “P. ostreatus + MWCNTs + lignin + glucose” and “P. ostreatus + MWCNT + lignin”, respectively. However, in the manuscript text it has been stated that “Interestingly, in the treatment with P. ostreatus or P. chrysogenum grown on lignin + glucose + MWCTNs, the major weight loss in sample was observed, suggesting a significant loss of MWCNTs.” This appears incorrect based on the results presented in Figure 6. The conclusion from Figure 6 would be that P. ostreatus degraded MWCNTs significantly more efficiently than P. chrysogenum. This appears to be reflected also in the toxicity to C. elegans. The respective samples where TGA indicated almost no residues of MWCNTs induced lower mortality in C. elegans compared to the respective MWCNT samples from P. chrysogenum samples (Figure 7). If this is correct, please modify the interpretation of the data correspondingly.

Another aspect that has not been discussed in the manuscript is the significantly higher mortality of C. elegans induced by the MWCNTs +P. ostreatus or P. chrysogenum treatment (Figure 7). The results of the enzymatic activities of the fungi, Raman, FTIR and TGA all appear to indicate that the highest level of MWCNT transformation/biodegradation resulted in the fungal cultures supplemented with lignin (with or without glucose). Still, the highest C. elegans mortality was induced by the fungi-treated MWCNTs (without the addition of lignin or glucose). Unfortunately, there are no TGA or FTIR characterizations of fungi + MWCNTs, but Raman analysis indicates no transformation of MWCNTs in P. chrysogenum cultures. What could have caused such a high mortality of C. elegans by incinerated MWCNTs compared to the original unincinerated MWCNTs? Please elaborate on this aspect of the study. The latter aspect of comparing the toxicity of unincinerated MWCNTs to the incinerated MWCNTs raises an additional question of the impact of the 4-h incineration of MWCNTs at 300C. How did this treatment, which was intended for elimination of fungal biomass and added organic nutrients, affect the oxygen-containing functional groups of MWCNTs? Could the physicochemical properties of MWCNTs have been modified by this treatment and thus caused the increased toxicity to C. elegans (i.e., not caused by fungal degradation of MWCNTs but the experimental procedure)? Is there a control treatment that would prove otherwise? It appears from the figure caption (of Figure 7) that the Control treatments which contained MWCNTs without fungal treatments were not incinerated, thus are not appropriate controls for the incinerated MWCNTs.

Additionally, in Figure 7, among the control samples, “MWCNTs + glucose” treatment induced significantly higher mortality in C. elegans than “MWCNTs” whereas “MWCNTs +lignin + glucose” induced comparable mortality with “MWCNTs”. Please elaborate on this effect: why would the addition of glucose to MWCNTs increase toxicity to C. elegans while the addition of lignin to MWCNTs and glucose did not induce higher toxicity? What kind of physico-chemical or biological effects could be at play here?

Also, the manuscript should be proof-read to correct the language errors (e.g., in several places "life" should be "live" etc., there are also other language and typographical errors that should be corrected).

Reviewer 2 ·

Basic reporting

no comment

Experimental design

The necessary explanations and corrections were made in the text of the manuscript.

Validity of the findings

There are no comments on the revised manuscript.

Additional comments

I thank the authors for the attentive attitude to comments and questions. All necessary changes have been made to the manuscript. I believe that the revised manuscript can be accepted for publication.

·

Basic reporting

Overall, all the comments were clearly answered.

Experimental design

Some comments are well answered and others revised with detailed info.

Validity of the findings

Results and discussion were cleared revised.

Additional comments

The revised manuscript was overall well organized and easy to read. It can be published as it is.

---

## Round 0.3 · accepted · Accept

I have no further comments on the manuscript.